# Genes Associated with Muscle, Tendon and Ligament Injury Epidemiology in Women's Amateur Football Players

David Varillas-Delgado [1,2]

1   Exercise and Sport Science, Faculty of Health Sciences, Universidad Francisco de Vitoria, 28223 Pozuelo, Spain; david.varillas@ufv.es or d.varillas@sportnomics.es
2   SPORTNOMICS S.L., 28922 Madrid, Spain

**Abstract:** *Background*: There is a lack of specific genetic studies regarding injuries in women's football. However, different genetic factors have been associated with tendon/ligament injuries in women football players. The aim of the study was to examine the genotypic frequencies of genes associated with injury risk and epidemiology in women's amateur football players and the aetiology of injuries. *Methods*: In total, 168 women's amateur football players from football clubs in the Spanish second division league and Caucasian descent were enrolled in this prospective observational cross-sectional study. *AMPD1* (rs17602729), *ACE* (rs4646994), *ACTN3* (rs1815739), *CKM* (rs8111989) and *MLCK* (rs2849757 and rs2700352) polymorphisms were genotyped. The characteristics of 169 non-contact injuries during the 2022/2023 season were classified following the International Olympic Committee (IOC) Consensus Statement for reporting injuries as follows: musculoskeletal, tendon/ligament, injury setting; and severity. The disequilibria of polymorphisms were estimated using the Hardy–Weinberg Equilibrium (HWE). The characteristics of the injuries were recorded, and the genotype characteristics were analysed. The genotype frequencies of all polymorphisms were compared between non-injured and injured football players and injury aetiologies. *Results*: The *AMPD1* genotype distribution differed between tendon/ligament injured and non-injured ($p = 0.003$) with a higher frequency in the TT genotype and T allele. The genotype distribution was different for the *CKM* and *MLCK* c.37885C>A polymorphisms in training and match injuries ($p = 0.038$ and $p = 0.031$, respectively). In the *ACTN3* and *AMPD1* polymorphisms, the distribution of the TT genotype in both genes showed a higher frequency in severe injuries (all $p < 0.001$). *Conclusions*: Tendon/ligament injury epidemiology in women's amateur football players was associated especially with the TT genotype of the *AMPD1* gene. The TT genotype of the *AMPD1* and *ACTN3* genes was also associated with severity, and the *CKM* and *MLCK* polymorphisms were associated with injury settings.

**Keywords:** genetics; women football players; epidemiology; musculoskeletal; tendon/ligament

## 1. Introduction

Football is the most played sport in the world by children and adults with varying degrees of experience and skill [1]. Women players must repeatedly perform sudden accelerations and decelerations, rapid changes in direction, jumps and landings [2,3]. These high-intensity situations, together with frequent exposure to collisions and contact, leads to a marked increase in the risk of injury in this sport compared to individual sports [4,5], and football is one of the five sports in which players are most at risk of injury [2].

Women football players have unique morphological and body composition characteristics [6] with a smaller body size and lower body mass than men football players. However, they have a higher percentage of body fat, which contributes to a higher epidemiology of fracture injuries [7,8].

The risk of serious knee injuries (such as anterior cruciate ligament (ACL) rupture) is at least twice as high in women as in men, regardless of exposure or level of participation [9]; they also have a higher risk of concussions and [10,11] knee [12] and ankle injuries [13].

Training intensity, biomechanics and hormonal differences may also contribute to this epidemiology of injuries in women football players [14,15]. The injury rate among professional and amateur players may vary, showing in amateur players an incidence of 9.6/1000 h of exposure and in professional football players, that of 8.1/1000 h of exposure [16]. Exercise-based prevention programmes may reduce the risk of non-contact musculoskeletal injuries by 23% among football players [16,17].

In recent years, genetics, especially in single nucleotide polymorphism (SNP) studies, has provoked great interest in susceptibility to injury and in the characteristics of tissues that are more prone to injury, because these play such important roles [18–20]. Genetics play an important role in determining the structure and function of connective tissue such as muscles, tendons and ligaments. Several genetic polymorphisms can influence the elasticity, strength and resilience of these tissues, which in turn can increase susceptibility to injury [21]. Some of the genes associated with the increased risk of muscle, tendon and ligament injuries include those related to the composition of collagen and elastin, which are key structural proteins in these tissues, and genetic variants that affect the production or quality of collagen may increase vulnerability to tendon and ligament injuries, specially the *COL5A1* gene [22]. There are some important reasons to consider, like genetic variability, tissue structure and composition, tissue functional properties, inflammatory responses and tissue repair and hereditary risk factors [21,23].

Several studies have shown the association of genetics, especially in muscle injuries, focusing on the alpha-actinin 3 (*ACTN3*) and angiotensin-converting enzyme isoform 1 (*ACE1*) genes [20,24–26]. However, recent studies focusing on genetic scores in genes related to sport injuries have shown that the gene associated with muscle metabolism, adenosine monophosphate deaminase 1 (*AMPD1*), is mostly responsible for muscle injuries in men's professional football players [18] and men and women elite endurance athletes [27], discovering new associations in muscle injuries in football players in the creatine kinase muscle (*CKM*) gene [19], and the myosin light chain kinase (*MLCK*) gene (also known as *MYLK*) and muscle damage during sports practice in men and women endurance runners [28]. To date, there has been a lack of specific genetic studies regarding injuries in women's football compared to other sports or even in men's football. However, different genetic factors have been associated with knee injuries in women football players [29].

Therefore, the aim of the research was to study the genotypic frequencies of genes associated with injury risk and epidemiology in musculoskeletal and tendon/ligament injuries and the aetiology of injuries occurring in women's amateur football players. The hypothesis was based on previous studies that genetics is associated with the epidemiology of injury in women football players and could have a significant effect on musculoskeletal and tendon/ligament injuries, especially in *AMPD1* and *ACTN3* polymorphisms.

## 2. Materials and Methods

### 2.1. Participants

This epidemiological cross-sectional descriptive study analysed 168 Spanish women's amateur football players in the Spanish second division league during the 2022/2023 season. All of the women's amateur football players were of Spanish Caucasian descent (Caucasian descent for the population of ≥3 generations). The characteristics of this cohort are presented in Table 1.

The following inclusion criteria were established: (a) women´s amateur football players >18 years; (b) women's amateur football players with a contract with the first team of the football club; (c) players who had participated in training and matches throughout the season at the same football club; and (d) players who had performed regular exercise training of >1 h per session, ≥3 days per week during the previous 6 months. Exclusion criteria were (a) players with only contact injuries during the season; (b) disabling injuries for football training or matches in the 6 months prior to the start of the study; and (c) women's professional football players.

**Table 1.** Women's amateur football players' demographic characteristics.

|  |  | Total | Injured | Non-Injured | *p* Value |
|---|---|---|---|---|---|
|  | Age, mean (SD) | 23.26 (3.37) | 23.63 (3.31) | 22.43 (4.13) | 0.459 |
|  | Weight, mean (SD) | 60.78 (6.08) | 60.15 (5.88) | 61.62 (6.55) | 0.853 |
|  | Height, mean (SD) | 163.91 (5.18) | 164.15 (4.99) | 163.10 (5.46) | 0.673 |
|  | BMI, mean (SD) | 22.63 (5.03) | 22.15 (4.88) | 23.32 (5.31) | 0.771 |
|  | Goalkeeper, n (%) | 16 (9.5) | 7 (7.5) | 9 (12.0) |  |
| Position | Defenders, n (%) | 49 (29.2) | 28 (30.1) | 21 (28.0) | 0.239 |
|  | Midfields, n (%) | 67 (39.9) | 34 (36.5) | 33 (44.0) |  |
|  | Forwards, n (%) | 36 (21.4) | 24 (25.9) | 12 (16.0) |  |

SD: standard deviation.

Written informed consent was obtained from all subjects. The protocol of the study was approved by the Ethics Committee of the University Francisco de Vitoria (UFV 32/2020) and conducted in accordance with the Declaration of Helsinki for Human Research of 1964 (last modified in 2013). All of the women football players in the study were included with codification identifiers.

### 2.2. Deoxyribonucleic Acid (DNA) Sample and Genotyping

Samples were collected with SARSTED swabs by buccal smear and kept refrigerated until genotyping.

DNA extraction from the swabs was carried out through automatic extraction using QIACube equipment (QIAGEN, Venlo, The Netherlands), yielding a DNA concentration of 25–40 ng/mL, which was kept in a solution in a volume of 100 μL at $-20$ °C until genotyping.

*ACE* I/D (rs4646994), *ACTN3* c.1729C>T (rs1815739), *AMPD1* c.34C>T (rs17602729), *CKM* c.\*800A>G (rs8111989) and *MLCK* c.49C>T (rs2700352) and c.37885C>A (rs28497577) polymorphisms were genotyped using Single Nucleotide Primer Extension (SNPE) with a SNaPshot Multiplex Kit (Thermo Fisher Scientific, Waltham, MA, USA), with an analysis of the reaction result conducted using capillary electrophoresis fragments, in an ABI3500 unit (Applied Biosystems, Foster City, CA, USA) with a bioinformatic analysis performed using GeneMapper 5.0 software (Applied Biosystems, Foster City, CA, USA).

### 2.3. Injury Recording

This research prospectively included the records of all the players during the 2022/2023 season with injuries, and their characteristics. For each injury, the medical staff provided the date of injury, onset (acute or repetitive/overuse), injury setting (training or match), player position (goalkeeper or field player), injury location, type of injury (the specific injury diagnosis was also recorded), injury severity based on layoff time (0 days (when a player could not participate fully on the day of an injury but was available for full participation the next day), slight (1–3 days), mild (4–7 days), moderate (8–28 days), or severe (>28 days)) [30] of all non-contact injuries reported by the clubs' medical services. To be included in the analysis, the injury had to have been a consequence of exposure to football during training or competition. Injuries caused by a collision with another player or an object were excluded from the investigation as they are potentially not affected by the player's genotype. The injury collection followed the international consensus statement on procedures for the epidemiological studies of injuries in football recommended by the Fédération Internationale de Football Association (FIFA) and the Union of European Football Associations (UEFA) [31,32]. The definition of the International Olympic Committee (IOC) consensus statement was used to classify injury severity reporting data on the day of the injury and the day of returning to play.

*2.4. Statistical Analysis*

The statistical analysis was performed using the Statistical Package for the Social Sciences (SPSS), v.21.0 for Windows. (IBM Corp. Released 2012. IBM SPSS Statistics for Windows, Version 21.0. IBM Corp.: Armonk, NY, USA). The normality of each variable was initially tested using the Kolmogorov–Smirnov test. Descriptive statistics were calculated for each genotype. The genotype frequencies of all polymorphisms were compared between non-injured and injured football players and injury aetiologies, using a $\chi^2$ test. The disequilibria of SNPs were estimated using the Hardy–Weinberg Equilibrium (HWE). The characteristics of the injuries were recorded, and the genotype characteristics were analysed using the $\chi^2$ test. To determine what genotype was associated with an unexpected distribution, the standardised residuals were calculated based on the difference between the observed and the expected values. Briefly, within each injury variable, a genotype was considered to have a statistically different distribution from the expected value when its distribution was > or < the critical Z-score value (i.e., 1.96).

The significance level was set at $p < 0.050$.

## 3. Results

All the polymorphisms analysed met the HWE (all $p > 0.05$). The genotype frequencies for the six polymorphisms in the women's amateur football players are shown in Table 2.

**Table 2.** Genotype frequencies in women´s amateur football players.

| | Gene | Polymorphism | dbSNP | Genotype | Women Football Players, n (%) |
|---|---|---|---|---|---|
| *ACE* | Angiotensin-converting enzyme | I/D | rs4646994 | DD | 86 (51.2) |
| | | | | ID | 57 (33.9) |
| | | | | II | 25 (14.9) |
| *ACTN3* | alpha-actinin-3 | c.1729C>T | rs1815739 | CC | 47 (27.9) |
| | | | | CT | 96 (57.2) |
| | | | | TT | 25 (14.9) |
| *AMPD1* | Adenosine monophosphate deaminase 1 | c.34C>T | rs17602729 | CC | 125 (74.4) |
| | | | | CT | 29 (17.2) |
| | | | | TT | 14 (8.4) |
| *CKM* | Muscle-specific creatine kinase | c.*800A>G | rs8111989 | GG | 22 (13.1) |
| | | | | GA | 75 (44.6) |
| | | | | AA | 71 (42.3) |
| *MLCK* | Myosin light-chain kinase | c.49C>T | rs2700352 | CC | 111 (66.1) |
| | | | | CT | 50 (29.7) |
| | | | | TT | 7 (4.2) |
| | Myosin light-chain kinase | c.37885C>A | rs28497577 | AA | 7 (4.2) |
| | | | | CA | 18 (10.7) |
| | | | | CC | 143 (85.1) |

In total, 93 (55.4%) women's amateur football players suffered a non-contact injury during the season, for a total of 169. The remaining 75 players (44.6%) had no injuries during the season. The characteristics of the injuries recorded and analysed during the season are presented in Table 3.

**Table 3.** Injury characteristics in women´s amateur football players during the 2022/2023 season.

|  |  | Frequency, n (%) |
|---|---|---|
| Onset | Acute | 160 (94.7) |
|  | Repetitive/overuse | 9 (5.3) |
| Injury setting | Match | 99 (58.6) |
|  | Training | 70 (41.4) |
| Tissue affected | Bone | 3 (1.8) |
|  | Joints and ligaments | 92 (54.4) |
|  | Muscles and tendons | 73 (43.2) |
|  | Skin | 0 (0.0) |
|  | Nervous system | 1 (0.6) |
|  | Other | 0 (0.0) |
| Severity | Slight | 70 (41.4) |
|  | Mild | 26 (15.4) |
|  | Moderate | 44 (26.0) |
|  | Severe | 29 (17.2) |
| Type | Fracture | 0 (0.0) |
|  | Other bone injuries | 3 (1.8) |
|  | Luxation/subluxation | 9 (5.3) |
|  | Ligamentous distension | 55 (32.5) |
|  | Ligamentous tear | 15 (8.9) |
|  | Meniscus injury | 3 (1.8) |
|  | Chondral injury | 1 (0.6) |
|  | Synovitis | 0 (0.0) |
|  | Fasciitis | 0 (0.0) |
|  | Muscle tear | 47 (27.8) |
|  | Muscle contracture | 23 (13.6) |
|  | Tendon strain | 3 (1.8) |
|  | Tendon rupture | 0 (0.0) |
|  | Bursitis | 0 (0.0) |
|  | Tendinitis | 10 (5.9) |
|  | Hematoma contusion | 0 (0.0) |
|  | Abrasion | 0 (0.0) |
|  | Incised wound | 0 (0.0) |
|  | Concussion | 0 (0.0) |
|  | Dental injuries | 0 (0.0) |
|  | Nerve injury | 0 (0.0) |
|  | Other injuries | 0 (0.0) |
| Recurrency | No | 143 (84.6) |
|  | Yes | 26 (15.4) |

The genotypic distribution in the epidemiology of injured women's amateur football players compared to the non-injured players showed no statistical differences in any polymorphism (all $p > 0.050$) (Table 4).

**Table 4.** Genotype distribution in women's amateur football players for total injuries, musculoskeletal and tendon/ligament injuries.

| | Polymorphism | Genotype | Injured | | | | Musculoskeletal Injuries | | | | Tendon/Ligament Injuries | | | |
|---|---|---|---|---|---|---|---|---|---|---|---|---|---|---|
| | | | No, n (%) | Yes, n (%) | *p* Value | df | No, n (%) | Yes, n (%) | *p* Value | df | No, n (%) | Yes, n (%) | *p* Value | df |
| *ACE* | I/D | DD | 40 (53.3) | 46 (49.5) | 0.275 | 4 | 38 (38.4) | 35 (50.0) | 0.682 | 8 | 35 (45.4) | 35 (38.0) | 0.756 | 10 |
| | | ID | 30 (40.0) | 27 (29.0) | | | 42 (42.4) | 26 (37.1) | | | 31 (40.3) | 38 (41.3) | | |
| | | II | 5 (6.7) | 20 (21.5) | | | 19 (19.2) | 9 (12.9) | | | 11 (14.3) | 19 (20.7) | | |
| *ACTN3* | c.1729C>T | CC | 20 (26.7) | 27 (29.0) | 0.966 | 11 | 35 (35.4) | 22 (31.4) | 0.364 | 3 | 27 (35.1) | 23 (25.0) | 0.700 | 8 |
| | | CT | 43 (57.3) | 53 (57.0) | | | 53 (53.5) | 44 (62.8) | | | 41 (53.2) | 58 (63.1) | | |
| | | TT | 12 (16.0) | 13 (14.0) | | | 11 (11.1) | 4 (5.8) | | | 9 (11.8) | 11 (11.9) | | |
| *AMPD1* | c.34C>T | CC | 61 (81.3) | 64 (68.8) | 0.219 | 2 | 54 (54.5) | 26 (37.1) | 0.317 | 12 | 41 (53.2) [↑] | 4 (4.3) [↓] | 0.003 | 3 |
| | | CT | 14 (18.7) | 15 (16.1) | | | 28 (28.3) | 31 (44.3) | | | 25 (32.5) | 57 (62.0) | | |
| | | TT | 0 (0.0) | 14 (15.1) | | | 17 (17.2) | 13 (18.6) | | | 11 (14.3) [↓] | 31 (33.7) [↑] | | |
| *CKM* | c.*800A>G | GG | 10 (13.3) | 12 (12.9) | 0.895 | 9 | 17 (17.2) | 9 (12.9) | 0.425 | 8 | 13 (16.9) | 11 (12.0) | 0.234 | 10 |
| | | GA | 35 (46.7) | 40 (43.0) | | | 28 (28.3) | 31 (44.3) | | | 30 (39.0) | 23 (25.0) | | |
| | | AA | 30 (40.0) | 41 (44.1) | | | 54 (54.5) | 30 (42.8) | | | 34 (44.1) | 58 (63.0) | | |
| *MLCK* | c.49C>T | CC | 50 (66.7) | 61 (65.6) | 0.597 | 2 | 75 (75.7) | 57 (81.4) | 0.686 | 2 | 66 (85.7) | 61 (66.3) | 0.135 | 2 |
| | | CT | 25 (33.3) | 25 (26.9) | | | 21 (21.3) | 13 (18.6) | | | 11 (14.3) | 27 (29.4) | | |
| | | TT | 0 (0.0) | 7 (7.5) | | | 3 (3.0) | 0 (0.0) | | | 0 (0.0) | 4 (4.3) | | |
| | c.37885C>A | AA | 7 (9.3) | 0 (0.0) | 0.127 | 2 | 0 (0.0) | 3 (4.3) | 0.617 | 2 | 0 (0.0) | 2 (2.1) | 0.279 | 2 |
| | | CA | 5 (6.7) | 13 (14.0) | | | 24 (24.3) | 13 (18.6) | | | 14 (18.2) | 25 (27.2) | | |
| | | CC | 63 (84.0) | 80 (86.0) | | | 75 (75.7) | 54 (77.1) | | | 63 (81.8) | 65 (70.7) | | |

[↑] indicates that the frequency was above the expected value, calculated from the standardised residuals of a chi-square test; [↓] indicates that the frequency was below the expected value, calculated from the standardised residuals of a chi-square test; df, degrees of freedom.

### 3.1. Musculoskeletal Injuries

Of the total number of injuries, 70 (41.4%) were musculoskeletal in 46 women's amateur football players (Table 3). No differences were observed for all polymorphisms between the non-injured musculoskeletal and the injured ones (all $p > 0.050$) (Table 4).

### 3.2. Tendon/Ligament Injuries

In total, 43 (25.5%) women's amateur football players suffered a total of 92 (54.3%) tendon/ligament injuries.

The *AMPD1* genotype distribution differed between tendon/ligament injured and non-injured players ($p = 0.003$) with a significant higher frequency of the TT genotype in injured versus non-injured players (33.7% vs. 14.3%, respectively) and a lower frequency of the CC in injured versus non-injured players (4.3% vs. 53.2%) (Table 4).

### 3.3. Injury Settings

Of the 169 non-contact injuries analysed, 99 (58.6%) occurred in matches, while 70 (41.4%) occurred during training.

The genotype distribution was different for the *CKM* c.*800A>G polymorphism ($p = 0.038$), showing that the GG genotype had a higher frequency in training injuries compared to those observed in matches (29.2% vs. 6.1%, respectively) (Table 5). The *MLCK* c.37885C>A polymorphism showed a higher frequency of the CA genotype in players injured in matches than those injured during training (32.4% vs. 8.3%) ($p = 0.031$) (Table 5).

**Table 5.** Genotype distribution in women´s amateur football players regarding injury setting and severity.

| Polymorphism | Genotype | Injury Setting | | | | Severity | | | |
|---|---|---|---|---|---|---|---|---|---|
| | | Training, n (%) | Match, n (%) | *p* Value | df | No, n (%) | Yes, n (%) | *p* Value | df |
| *ACE* I/D | DD | 23 (32.8) | 47 (47.5) | | | 55 (39.3) | 14 (48.3) | | |
| | ID | 32 (45.7) | 37 (37.4) | 0.427 | 14 | 61 (43.6) | 9 (31.0) | 0.683 | 5 |
| | II | 15 (21.5) | 15 (15.1) | | | 24 (17.1) | 6 (20.7) | | |
| *ACTN3* c.1729C>T | CC | 23 (32.8) | 29 (29.3) | | | 50 (35.7) | 3 (10.3) | | |
| | CT | 38 (54.3) | 58 (58.6) | 0.937 | 8 | 85 (60.7) | 11 (37.9) | <0.001 | 4 |
| | TT | 9 (12.9) | 12 (12.1) | | | 5 (3.6) ↓ | 15 (51.8) ↑ | | |
| *AMPD1* c.34C>T | CC | 38 (54.3) | 46 (46.5) | | | 61 (43.6) ↑ | 1 (3.4) ↓ | | |
| | CT | 18 (25.7) | 38 (38.4) | 0.465 | 13 | 50 (35.7) | 6 (20.7) | <0.001 | 2 |
| | TT | 14 (20.0) | 15 (15.1) | | | 29 (20.7) ↓ | 22 (75.9) ↑ | | |
| *CKM* c.*800A>G | GG | 20 (29.2) ↑ | 6 (6.1) ↓ | | | 20 (14.6) | 5 (17.3) | | |
| | GA | 23 (33.3) | 32 (32.3) | 0.038 | 5 | 38 (27.1) | 13 (44.8) | 0.199 | 4 |
| | AA | 27 (37.5) | 61 (61.6) | | | 82 (58.3) | 11 (37.9) | | |
| *MLCK* c.49C>T | CC | 52 (74.3) | 75 (75.8) | | | 114 (81.4) | 17 (58.6) | | |
| | CT | 15 (21.4) | 20 (20.2) | 0.843 | 2 | 23 (16.4) | 12 (41.4) | 0.238 | 2 |
| | TT | 3 (4.3) | 4 (4.0) | | | 3 (2.1) | 0 (0.0) | | |
| c.37885C>A | AA | 1 (1.4) | 2 (2.0) | | | 2 (1.4) | 2 (6.7) | | |
| | CA | 5 (7.1) ↓ | 30 (30.3) ↑ | 0.024 | 2 | 33 (23.6) | 9 (31.0) | 0.301 | 2 |
| | CC | 64 (91.5) | 67 (67.7) | | | 105 (75.0) | 18 (62.3) | | |

↑ indicates that the frequency was above the expected value, calculated from the standardised residuals of a chi-square test; ↓ indicates that the frequency was below the expected value, calculated from the standardised residuals of a chi-square test; df, degrees of freedom.

### 3.4. Severity

In total, 29 (17.1%) injuries were severe, while 140 (82.9%) were non-severe.

In the *ACTN3* c.1729C>T polymorphism, the genotype distribution differed between severely injured and non-severely injured players ($p < 0.001$), showing in the TT genotype a higher frequency in players with severe injuries than those with non-severe injuries (51.8%

vs. 3.6%) (Table 5). Regarding the *AMPD1* c.34C>T polymorphism, differences in genotype distribution were observed in the TT genotype, with a higher frequency in players with severe injuries than those with non-severe injuries (75.9% vs. 20.7%) ($p < 0.001$) (Table 5).

## 4. Discussion

Although the physiology and aetiology of injuries in the world of women's football have been studied extensively, genetics remains a field that deserves more interest, and the author is presenting, for the first time, a study on the association between muscle injury-related genes and the epidemiology of injuries in women's amateur football players during a whole season. The aim of this study was to examine the association between muscle injury-related genes, injury risk and injury aetiology in women's amateur football players. The main finding is the correlation between genotype frequencies and different injury characteristics, with the C allele of the *AMPD1* c.34C>T polymorphism, suggesting some kind of benefit related to injury prevention, especially for protection against tendon/ligament injuries and severity, as well as with the C allele of the *ACTN3* c.1729C>T polymorphism.

The morphological characteristics of women football players may increase the risk of injuries such as stress fractures, and affect their overall strength and power on the pitch, as well as expose them to higher risks to certain types of injuries, as shown in previous studies on the Spanish League [15,33].

In recent years, the demands of training and matches have increased considerably in women's football, even for amateurs [34]. The overall epidemiology of injuries is similar to that of men's football, although the proportion of severe injuries is higher in women football players [2,35], totalling 17.2% in this study, showing a higher frequency of injuries than previously reported in men football players [36], which could be associated with significant costs [37]. The risk of severe knee injuries and ankle sprains is at least twice as high in women as in men football players, regardless of exposure or level of participation [38,39], which is consistent with the data presented in this study.

Genetics may influence the risk of muscle injury in women football players, with a focus on tendon and ligament injuries. A recent study found that variations in the *COL5A1* rs13946 polymorphism were associated with increased risks of ACL tears and muscle injuries in women's football players. These variations were associated with differences in ligament laxity between genders, making women football players more susceptible to these injuries [22].

In this regard, the *AMPD1* c.34C>T polymorphism in this cohort of women's amateur football players showed different injury aetiology characteristics from previous studies in men's professional football players [18]. The CC genotype was more frequent in non-injured players (53.2%) compared to players with the TT genotype (14.3%), an aspect that could present the CC genotype as a protector genotype in tendon and ligament injuries. This is the first time that these results have been presented for women's football, also showing an association with the severity of injuries in the TT genotype (75.9%), as has been presented in men's professional football players [18].

Other studies have found that variations in the *ACTN3* gene are associated with an increased risk of muscle injury. Specifically, individuals with the TT genotype in the *ACTN3* c.1729C>T polymorphism could have an increased risk of muscle strains and tears, which are common injuries in women football players [25,40]. The results presented in this study show a lower severity of injuries in the CC genotype compared to the TT genotype, similar to the data found previously in men's professional football players [18].

There was an association between match and training injuries in muscle damage genes, such as the *CKM* c.*800A>G polymorphism, in which, when comparing training and match injuries, it was shown that the GG genotype and G allele lead to higher risks of training injuries in women's amateur football players. These data differ from those found in a recent study in which no association was found between this polymorphism and the risk of training and match injuries in men's professional football players [19]. To date, the *CKM* gene has not been extensively studied in relation to muscle injury, but it has been shown

that it can be used for athlete selection due to the effect that the GG genotype has on muscle damage in football players [18,41].

One study found that variations in the *MLCK* c.37885C>A polymorphism are associated with an increased risk of muscle strains in professional football players [18]. However, this study has shown that this polymorphism, specifically the CA genotype, is associated with match injuries, with the CC genotype being protective for these injury settings, similar to those found in a recent study on men football players in which the CA genotype had a higher risk of injury in matches versus training [18]. The exact mechanisms by which variations in the *MLCK* gene may influence injury risk are not yet fully understood, and further research is needed to confirm the results found. However, it is thought that these variations may affect the function of smooth muscle cells in the body, including those found in the muscles and connective tissues that are most vulnerable to injury in football [42].

This investigation shows, for the first time, that genetics may be a significant factor in the predisposition to increase the risk of epidemiology in the injuries of women's amateur football players, especially tendon and ligament injuries, showing new candidate genes associated with the aetiology of these injuries: *AMPD1* and *ACTN3*.

However, this study presents several limitations: (a) the amateur standard of football players recruited for this study may well have confounded these results, as the injuries may have been influenced by lower levels of strength/fitness and/or lower levels of scientific support in this population. Thus, the small effect of individual genetic polymorphisms on injury risk (which is multifactorial even in professional football players) is more likely to be elucidated in a higher standard of football players. (b) Given the small effect size of an individual genetic polymorphism and the multifactorial nature of injury aetiology, the sample size is too low to provide any meaningful information regarding a genetic association with injury. In the future, the author will collect more data by significantly increasing the sample size and conducting a follow-up. (c) Training methods and injury prevention programmes may condition injury risk and be a confounding factor for this study. Finally, (d) future studies could include epigenetic and environmental aspects in the analysis of factors associated with women's football performance, such as menstrual cycle, to complement the genetic analyses and their applicability in women's professional football players and to improve and strengthen the links between genetic variations and the knowledge of injuries in this cohort of women football players.

## 5. Conclusions

This investigation is the first to demonstrate that injury risk and epidemiology in women's amateur football players is associated with a combination of related genes in muscle injuries. Especially, the TT genotype and the T allele of the *AMPD1* c.34C>T polymorphism appears to be the polymorphism that is best associated with tendon/ligament injuries and severity in this cohort of women football players. It has also been shown that the TT genotype of the *ACTN3* c.1729C>T polymorphism is associated with injury severity, while the muscle damage-related genes, *CKM* and *MLCK*, are associated with injury setting.

The outcomes of this research suggest that the addition of genetic variants previously associated with muscle injury in men football players may serve as a predictor of both muscle and tendon/ligament injury in women football players for the medical and fitness staff. These outcomes also suggest the need to replicate the findings in genetic studies on the aetiology of these injuries in women's professional/amateur football players.

**Funding:** This research was funded by the MAPFRE Foundation, Ignacio H. de Larramendi, grant number 6391.

**Institutional Review Board Statement:** The study was conducted in accordance with the Declaration of Helsinki and approved by the Ethics Committee of Universidad Francisco de Vitoria (protocol code: 32/2020).

**Informed Consent Statement:** Informed consent was obtained from all subjects involved in the study. Written informed consent was obtained from the subjects to publish this paper.

**Data Availability Statement:** The raw data supporting the conclusions of this article will be made available by the authors on request.

**Acknowledgments:** The author is grateful for the participation of the women football players in this research and wishes to thank the medical staff for their invaluable contribution to the study.

**Conflicts of Interest:** Author David Varillas-Delgado was employed by the company SPORTNOMICS S.L. Author declare that the research was conducted in the absence of any commercial or financial relationships that could be construed as a potential conflict of interest.

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
