# Peer review of "Genes Associated with Muscle, Tendon and Ligament Injury Epidemiology in Women’s Amateur Football Players"

_applsci, doi:10.3390/app14051980_

Round 1

Reviewer 1 Report

Comments and Suggestions for Authors

The manuscript is clear, relevant to the field and presented in a well-organised manner. The references cited are not recent publications (within the last 5 years). The manuscript is scientifically sound and the experiment is adequate to test the hypothesis. The results of the manuscript are reproducible based on the details given in the methods section. Tables are appropriate. They present the data correctly. They are easy to interpret and understand. Data are interpreted appropriately and consistently throughout the manuscript. Statistical analysis is correct.

Author Response

Response to Reviewers’ comments

applsci-2878059

Genes associated with muscle, tendon and ligament injury epidemiology in women’s amateur football players.

The author sincerely thank the Reviewer for carefully proofreading the manuscript and for their helpful and constructive comments. I have addressed all the points raised by the Reviewer in this response letter and we have highlighted any change to manuscript in red using the track changes tool. I believe that the manuscript has been improved by the suggested changes.

Reviewer: 1

The manuscript is clear, relevant to the field and presented in a well-organised manner. The references cited are not recent publications (within the last 5 years). The manuscript is scientifically sound and the experiment is adequate to test the hypothesis. The results of the manuscript are reproducible based on the details given in the methods section. Tables are appropriate. They present the data correctly. They are easy to interpret and understand. Data are interpreted appropriately and consistently throughout the manuscript. Statistical analysis is correct.

Thank for your positive comments and the valuable review of this manuscript.

On behalf of all co-authors, many thanks for the insightful review.

Reviewer 2 Report

Comments and Suggestions for Authors

I read this paper with great interest and congratulate the authors for conducting such an extensive study on such a large group of elite athletes. I am also impressed by such meticulously planned methods and presented research results.

The manuscript, titled: "Genes associated with muscle, tendon and ligament injury epidemiology in women's amateur soccer players," points out the glaring lack of detailed genetic studies of injury in women's soccer compared to other sports and even men's soccer, and highlights the important role of conducting them and the need to repeat and expand them.

Introduction:

The introduction is complete and makes the rationale for this type of research very clear. It points to the need for a broader view of the problem of serious injury risk in female soccer players, where they are shown to be at least twice as common in women as in men, regardless of exposure or level of participation; they also have a higher risk of suffering concussions, knee and ankle injuries than male soccer players. The authors also point to issues of training intensity, biomechanics and hormonal differences that may also contribute to the higher injury epidemiology in female soccer players. The complexity and multifaceted nature of this problem is detailed.

Material and methods:

The material and methods are very well suited to the research problem. It was presented in a detailed and clear manner. The surveys used were selected as relevant to the research problem as possible, and the sampling of such a large group allowed for reliable data analysis.

Results:

The results presented in a reliable and convincing manner. Presenting them in the form of tables allowed a very clear analysis of the information contained in them.

Discussion and Conclusions:

The discussion and conclusions are well-written, and include a reference of the results of this work to those of other authors. The authors, through their research, succeeded in identifying an important area of existing factors and variables related to training and match injuries versus genetics in female soccer players.

References:

Literature review prepared quite exhaustively.

Author Response

Response to Reviewers’ comments

applsci-2878059

Genes associated with muscle, tendon and ligament injury epidemiology in women’s amateur football players.

The author sincerely thank the Reviewer for carefully proofreading the manuscript and for their helpful and constructive comments. I have addressed all the points raised by the Reviewer in this response letter and we have highlighted any change to manuscript in red using the track changes tool. I believe that the manuscript has been improved by the suggested changes.

Reviewer: 2

I read this paper with great interest and congratulate the authors for conducting such an extensive study on such a large group of elite athletes. I am also impressed by such meticulously planned methods and presented research results.

Thank for your positive comments and the valuable review of this manuscript.

The manuscript, titled: "Genes associated with muscle, tendon and ligament injury epidemiology in women's amateur soccer players," points out the glaring lack of detailed genetic studies of injury in women's soccer compared to other sports and even men's soccer, and highlights the important role of conducting them and the need to repeat and expand them.

Introduction:

The introduction is complete and makes the rationale for this type of research very clear. It points to the need for a broader view of the problem of serious injury risk in female soccer players, where they are shown to be at least twice as common in women as in men, regardless of exposure or level of participation; they also have a higher risk of suffering concussions, knee and ankle injuries than male soccer players. The authors also point to issues of training intensity, biomechanics and hormonal differences that may also contribute to the higher injury epidemiology in female soccer players. The complexity and multifaceted nature of this problem is detailed.

Thanks for this positive comments.

Material and methods:

The material and methods are very well suited to the research problem. It was presented in a detailed and clear manner. The surveys used were selected as relevant to the research problem as possible, and the sampling of such a large group allowed for reliable data analysis.

Thanks for this positive comments.

Results:

The results presented in a reliable and convincing manner. Presenting them in the form of tables allowed a very clear analysis of the information contained in them.

Thanks for this positive comments.

Discussion and Conclusions:

The discussion and conclusions are well-written, and include a reference of the results of this work to those of other authors. The authors, through their research, succeeded in identifying an important area of existing factors and variables related to training and match injuries versus genetics in female soccer players.

Thanks for this positive comments.

References:

Literature review prepared quite exhaustively.

Thanks for this positive comments.

On behalf of all co-authors, many thanks for the insightful review.

Reviewer 3 Report

Comments and Suggestions for Authors

Congratulations for your investigation. The following comments could improve your article:

The exploration of genetic factors in injury prevention within the context of women's soccer presents an intriguing avenue for analysis. I am particularly intrigued by the potential implications of your findings for extrapolation to other sporting disciplines. Have you considered the extent to which your research outcomes could be applicable across different sports?

Furthermore, I believe there may be opportunities to enhance the comprehensiveness of your study. Would it be feasible to extend the duration of follow-up and expand the sample size to bolster the robustness of your findings?

Given the prevalence of injuries observed in your study, I am curious to know if you have developed any specific prevention programs tailored to address these issues. Incorporating insights from your research into targeted injury prevention initiatives could have profound implications for athlete well-being and performance outcomes.

I would appreciate the inclusion of that content in the final article.

Author Response

Response to Reviewers’ comments

applsci-2878059

Genes associated with muscle, tendon and ligament injury epidemiology in women’s amateur football players.

The author sincerely thank the Reviewer for carefully proofreading the manuscript and for their helpful and constructive comments. I have addressed all the points raised by the Reviewer in this response letter and we have highlighted any change to manuscript in red using the track changes tool. I believe that the manuscript has been improved by the suggested changes.

Reviewer: 3

Congratulations for your investigation. The following comments could improve your article:

Thank for your positive comments and the valuable review of this manuscript.

The exploration of genetic factors in injury prevention within the context of women's soccer presents an intriguing avenue for analysis. I am particularly intrigued by the potential implications of your findings for extrapolation to other sporting disciplines. Have you considered the extent to which your research outcomes could be applicable across different sports?

Thank you for your suggestion. In this case, the author has been involved in research in other sports, such as elite athletes and male professional players as present in lines 71-72 with a new reference to support this previous research in other sporting disciplines (https://pubmed.ncbi.nlm.nih.gov/35921847/), being in these cohorts of professional athletes difficult to be extrapolated directly with the results obtained in amateur women's football players.

Furthermore, I believe there may be opportunities to enhance the comprehensiveness of your study. Would it be feasible to extend the duration of follow-up and expand the sample size to bolster the robustness of your findings?

I appreciate that point of view. In that regard, the author is conducting future studies with a longer follow-up period, as marked in limitations, adding the limitation of follow-up in future studies (line 355).

I would appreciate the inclusion of that content in the final article.

I hope I have been able to respond to the reviewer's considerations and increase the robustness of the study along with other reviewers' revisions in the final manuscript.

On behalf of all co-authors, many thanks for the insightful review.

Reviewer 4 Report

Comments and Suggestions for Authors

Comments are collected in notes format in the document. Two fundamental aspects must be considered. Sample calculation and collection of lesions following Fuller's model and used in previous studies of the same population. 10.3390/ijerph18063009

Furthermore, given that it is amateur soccer, it is contradictory that they have medical services. This should be clarified. Finally, the continued comparison with men, where this study has not examined these, is a mistake in my opinion.

Comments on the Quality of English Language

Oxford comma or some expressions need to be corrected.

Author Response

Response to Reviewers’ comments

applsci-2878059

Genes associated with muscle, tendon and ligament injury epidemiology in women’s amateur football players.

The author sincerely thank the Reviewer for carefully proofreading the manuscript and for their helpful and constructive comments. I have addressed all the points raised by the Reviewer in this response letter and we have highlighted any change to manuscript in red using the track changes tool. I believe that the manuscript has been improved by the suggested changes.

Reviewer: 4

Comments are collected in notes format in the document. Two fundamental aspects must be considered. Sample calculation and collection of lesions following Fuller's model and used in previous studies of the same population. 10.3390/ijerph18063009

The author has responded to the reviewer's considerations in the submitted paper and has marked improvements in the final manuscript.

Furthermore, given that it is amateur soccer, it is contradictory that they have medical services. This should be clarified. Finally, the continued comparison with men, where this study has not examined these, is a mistake in my opinion.

Amateur football clubs are women's clubs within the organization chart of men's teams or affiliates of women's professional teams where medical services are mandatory and necessary. The comparison with men has been deleted throughout the manuscript.

Oxford comma or some expressions need to be corrected.

Thanks for these suggestions. Oxford comma and several expressions has been corrected.

On behalf of all co-authors, many thanks for the insightful review.

Reviewer 5 Report

Comments and Suggestions for Authors

Abstract: The Introduction should include the statistical analysis performed. 

In lines 47-49, expand on the references to statements made in relation to the intensities of training and explain the differences.

In lines 50-52, cite studies that corroborate the statements made. 

Material and Methods: 

Please provide a more detailed explanation of the injured and non-injured players, including their distribution and whether one of the samples is overrepresented. Additionally, please clarify if there are any underage players and if so, whether their parents have provided informed and signed consent. 

Lines 133: Please explain whether the sample follows a normal distribution or not. 

Results: 

Line 76: Line 139 and onwards: For example, goalkeepers may experience more injuries due to the large amount of contact with the ground caused by their own actions. The study should explain the results based on playing positions, as this can be an important factor. Additionally, the study should investigate whether BMI affects player injuries, as this variable can substantially impact injury rates. 

In lines 265-269 of the discussion, please provide an explanation for the assertions made and the studies or results on which they are based.

In lines 265-269 of the discussion, please provide an explanation for the assertions made and the studies or results on which they are based. In lines 265-269 of the discussion, please provide an explanation for the assertions made and the studies or results on which they are based. It is important to avoid making assertions that are not supported by your research. 

Line 274: It is not entirely correct to say that women's football has been professional in recent years, as it has only been professional since 2023. Additionally, in your study, you refer to second division players who are not considered professionals today. Therefore, it is recommended to refer to them as amateurs.

Author Response

Response to Reviewers’ comments

applsci-2878059

Genes associated with muscle, tendon and ligament injury epidemiology in women’s amateur football players.

The author sincerely thank the Reviewer for carefully proofreading the manuscript and for their helpful and constructive comments. I have addressed all the points raised by the Reviewer in this response letter and we have highlighted any change to manuscript in red using the track changes tool. I believe that the manuscript has been improved by the suggested changes.

Reviewer: 5

Abstract: The Introduction should include the statistical analysis performed.

Thanks for this suggestion. Statistical analysis has been added in abstract. Lines 19-24.

In lines 47-49, expand on the references to statements made in relation to the intensities of training and explain the differences.

Thank you for your suggestion. In agreement with other reviewers, the differences in injuries between women and men have been removed and more references have been added to explain training intensities and methods to prevent injuries in football players, as indicated on lines 48-52.

In lines 50-52, cite studies that corroborate the statements made.

Thank you for your suggestion. Added 3 new recent citations to support this statement and substantiate the recent importance of genetics in the study of football injuries.

Material and Methods:

Please provide a more detailed explanation of the injured and non-injured players, including their distribution and whether one of the samples is overrepresented. Additionally, please clarify if there are any underage players and if so, whether their parents have provided informed and signed consent.

Thank you for your suggestion. As noted by the author, there has been no overrepresentation of injured and non-injured players, showed in results; lines 187-190.

All women´s amateur football players were over 18 years old and have been marked as inclusion criteria (lines 97-98).

Lines 133: Please explain whether the sample follows a normal distribution or not.

Thanks for this suggestion. The normality analysis has been added in Statistical Analysis subsection. Lines 144-145.

Results:

Line 76: Line 139 and onwards: For example, goalkeepers may experience more injuries due to the large amount of contact with the ground caused by their own actions. The study should explain the results based on playing positions, as this can be an important factor. Additionally, the study should investigate whether BMI affects player injuries, as this variable can substantially impact injury rates.

Thanks for the suggestion.  In this case, the target is genetics in tendon and ligament injuries. As shown in Table 1, there are no differences in the positions of the players and BMI also does not influence the incidence of injuries in amateur female football players. The injury rates has not been performed, because the data on training and match hours have not been reported due to the non-professionalization of the teams included in the study.

In lines 265-269 of the discussion, please provide an explanation for the assertions made and the studies or results on which they are based. It is important to avoid making assertions that are not supported by your research.

Thanks for this suggestion. The author has deleted this paragraph due to these assertions are not supported by research and are not compared to male football players.

Line 274: It is not entirely correct to say that women's football has been professional in recent years, as it has only been professional since 2023. Additionally, in your study, you refer to second division players who are not considered professionals today. Therefore, it is recommended to refer to them as amateurs.

Thanks for this suggestion. According to expert reviewer consideration, this sentence has been modified; line 297.

On behalf of all co-authors, many thanks for the insightful review.

Reviewer 6 Report

Comments and Suggestions for Authors

I appreciate the opportunity to review this exciting report about genes associated with muscle, tendon, and ligament injury epidemiology in women’s amateur football players, in which the author has concluded from the binary logistic models regressed on data provided through a cross-sectional design. Even though I find this manuscript interesting, some severe questions arose upon the reading, which I will address below point-by-point.

Lines 32-47: I find the gender comparison out of the scope of this research since the study examined genes only in the sample of women and made no comparisons with men. I find an abundance of reasons why this study should have been conducted, such as prevention of injury. Please revise the rationale.    

Lines 50-52: Please provide a reference.

Lines 50-55: Please explain how muscle, tendon, and ligament injury-associated genes prone individuals to increased injury risk.

Lines 56-66: Why are those specific genes selected? How do the genes increase the risk of muscle, tendon, and ligament injuries?

Lines 63-66: The study has no data on gender gene comparison.

Line 75: I find this research to have a cross-sectional design. The author collected all variables at once and did not track variables over time to detect an incidence.

Line 75: Has the author estimated the minimal sample size required? Was any randomization performed?

Lines 126-132: Why did the author use a χ2 test? To what research question should the response be?

133-137: Please report whether the assumptions are valid for applying the binary logistic regression model.

133-137: Please write what model measures are reported and how the effect size is interpreted.

Line 139: Please include χ2 value and df throughout the result section.

Lines 172-182: Also, I find that insufficient data from a binary logistic regression model have been reported (e.g., Nagelkerke R2 is missing) and poorly interpreted (e.g., who has more chance, how much…).

Finally, the review stops here due to the urgent need for subsequent changes to be made by the author.

Author Response

Response to Reviewers’ comments

applsci-2878059

Genes associated with muscle, tendon and ligament injury epidemiology in women’s amateur football players.

The author sincerely thank the Reviewer for carefully proofreading the manuscript and for their helpful and constructive comments. I have addressed all the points raised by the Reviewer in this response letter and we have highlighted any change to manuscript in red using the track changes tool. I believe that the manuscript has been improved by the suggested changes.

Reviewer: 6

I appreciate the opportunity to review this exciting report about genes associated with muscle, tendon, and ligament injury epidemiology in women’s amateur football players, in which the author has concluded from the binary logistic models regressed on data provided through a cross-sectional design. Even though I find this manuscript interesting, some severe questions arose upon the reading, which I will address below point-by-point.

Thanks for this positive comments. The author has responded point by point to the reviewer's expert considerations in order to improve the quality of the manuscript.

Lines 32-47: I find the gender comparison out of the scope of this research since the study examined genes only in the sample of women and made no comparisons with men. I find an abundance of reasons why this study should have been conducted, such as prevention of injury. Please revise the rationale.   

Thank you for this suggestion. Based on other considerations by other reviewers, the comparison with male football players has been removed from the manuscript.

Lines 50-52: Please provide a reference.

References has been provided.

Lines 50-55: Please explain how muscle, tendon, and ligament injury-associated genes prone individuals to increased injury risk.

Thanks for this suggestion. Has been added a paragraph explaining how muscle, tendon and ligament-associated genes prone to increase injury risk. Lines 55-63.

In lines 66-78 in turn explains several genes associated with previously identified muscle injuries.

Lines 56-66: Why are those specific genes selected? How do the genes increase the risk of muscle, tendon, and ligament injuries?

These genes have been selected for previous association in muscle injuries. Due to the association found, it is hypothesized that they may have causality with tendon and ligament injuries, as presented in lines 81-84.

Lines 63-66: The study has no data on gender gene comparison.

Thank you for this appreciation. Based on another reviewer's suggestion, the author has added a reference showing this same genetic profile in male and female runners (https://pubmed.ncbi.nlm.nih.gov/35921847/) for completeness.

The scarcity of research on female athletes and genetics means that there are not many studies with comparisons between sexes in the selected genes, which is another reason for the hypotheses put forward to increase knowledge in this field.

Line 75: I find this research to have a cross-sectional design. The author collected all variables at once and did not track variables over time to detect an incidence.

I appreciate the reviewer's consideration. Indeed, the study is cross-sectional and has been modified in the study design throughout the manuscript for better understanding.

Line 75: Has the author estimated the minimal sample size required? Was any randomization performed?

Thank you for your suggestion. The sample size calculation has not been estimated due to the voluntary participation of the players and based on other previous studies with a similar number and in which the sample size calculation has not been performed. The study is not randomized.

Lines 126-132: Why did the author use a χ2 test? To what research question should the response be?

The author has used the chi square test (χ2 test) due to the qualitative nature of the genetic variables analyzed in the study.

133-137: Please report whether the assumptions are valid for applying the binary logistic regression model.

Thank you for this consideration. Indeed, the reviewer is right that binary logistic regression is not valid to apply in the manuscript because binary data are not analyzed.

In this aspect, the author has eliminated the reported analysis and removed it from the results for better understanding.

133-137: Please write what model measures are reported and how the effect size is interpreted.

In this study, it is noted that the differences in genotypes were analyzed using the corrected typified waste as showed in lines 152-154;Briefly, within each injury variable, a genotype was considered to have a statistically different distribution from the expected value when its distribution was > or < the critical Z-score value (i.e., 1.96)”.

With respect to the reports of other reviewers, the statistics performed for the analysis of genotypes is correct and in accordance with the objectives set.

Line 139: Please include χ2 value and df throughout the result section.

Thanks for this suggestion. df has been added in table 4 and 5 for a better understanding of results.

Lines 172-182: Also, I find that insufficient data from a binary logistic regression model have been reported (e.g., Nagelkerke R2 is missing) and poorly interpreted (e.g., who has more chance, how much…).

Thanks for this suggestion. Binary logistic regression has been deleted and this consideration has been solvent previously.

Finally, the review stops here due to the urgent need for subsequent changes to be made by the author.

Thank you for your review and I hope that the considerations reported were the right ones to continue revising and improving the manuscript.

On behalf of all co-authors, many thanks for the insightful review.

Round 2

Reviewer 4 Report

Comments and Suggestions for Authors

Congratulations